# The Difference of Zagreb Indices of Halin Graphs

**Lina Zheng** [1], **Yiqiao Wang** [2] and **Weifan Wang** [1,*]

1　School of Mathematical Sciences, Zhejiang Normal University, Jinhua 321004, China; lnzheng@zjnu.cn
2　School of Management, Beijing University of Chinese Medicine, Beijing 100029, China
*　Correspondence: wwf@zjnu.cn

**Abstract:** The difference of Zagreb indices of a graph $G$ is defined as $\Delta \mathrm{M}(G) = \sum\limits_{u \in V(G)} (d(u))^2 - \sum\limits_{uv \in E(G)} d(u)d(v)$, where $d(x)$ denotes the degree of a vertex $x$ in $G$. A Halin graph $G$ is a graph that results from a plane tree $T$ without vertices of degree two and with at least one vertex of degree at least three such that all leaves are joined through a cycle $C$ in the embedded order. In this paper, we establish both lower and upper bounds on the difference of Zagreb indices for general Halin graphs and some special Halin graphs with fewer inner vertices. Furthermore, extremal graphs attaining related bounds are found.

**Keywords:** difference of Zagreb indices; Halin graphs; extremal graphs

**MSC:** 05C10

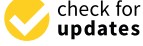



## 1. Introduction

All graphs that we consider in this paper are simple. If $G$ is a graph, then we use $V(G)$, $E(G)$, $\Delta(G)$, and $\delta(G)$ to denote its vertex set, edge set, maximum degree, and minimum degree, respectively. Let $n = |V(G)|$ and $m = |E(G)|$. For a vertex $y \in V(G)$, we denote by $d_G(y)$ the degree of $y$ in $G$ (in short, $d(y)$).

A topological index is indeed a quantity associated with chemical composition, which reveals a close connection between chemical structure and many physical properties, chemical reactivity, or biological activity. When we discuss a topological index that results from the vertex degree of a graph, we call it a degree-based index. Given a graph $G$, we define the first Zagreb index $\mathrm{M}_1(G)$ and the second Zagreb index $\mathrm{M}_2(G)$ as follows:

$$\mathrm{M}_1(G) = \sum_{u \in V(G)} (d(u))^2, \quad \mathrm{M}_2(G) = \sum_{uv \in E(G)} d(u)d(v).$$

The parameter $\mathrm{M}_1(G)$ was known to appear in some approximate description regarding the total $\pi$-electron energy [1] in 1972, and Gutman et al. introduced the parameter $\mathrm{M}_2(G)$ to measure the branching of the carbon atom skeleton [2] in 1975. The first employment of the name Zagreb indices appeared in a survey article [3]. A wealth of results above these two indices in graph theory was already collected in the survey article [4]. Recently, Pei and Pan [5] established some upper bounds for the Zagreb indices of trees in which distance $k$-domination number were given, and, moreover, they developed a characterization of extremal trees. Das and Ali [6] provided maximum value on the second Zagreb index for all connected graphs according to its order and cyclomatic number. Patil and Yattinahalli [7] obtained explicit formulae for the second Zagreb index of some special graphs such as semitotal-line graphs, semitotal-point graphs, and total transformation graphs. Yang and Deng [8] determined maximal values and maximal graphs for the first Zagreb index of unicyclic digraphs with respect to order and matching number.

Let $\Delta M(G) = M_2(G) - M_1(G)$, which is said to be the difference of Zagreb indices of a graph $G$ (see [9,10]). Using a quick computation, we may easily deduce the following meaningful expression:

$$\Delta M(G) = M_2(G) - M_1(G) = \sum_{uv \in E(G)} (d(u) - 1)(d(v) - 1) - m.$$

More recently, many interesting results about the difference of Zagreb indices of given graphs were obtained. According to the order and cyclicity of $G$, Caporossi et al. [11] provided two nice lower bounds on $\Delta M(G)$ for representation. Milošević et al. [12] researched into the graphs, where $\Delta M(G)$ became an integer. Furtula et al. [10] gave many fundamental features of $\Delta M(G)$. A more thorough characterization of graphs depending on the size of $\Delta M(G)$ was set up in [13]. Wang and Yuan [14] discussed the difference between $M_1(G)$ and $M_2(G)$ to yield some interesting extremal results and related structural properties. Horoldagva et al. [15] investigated the cyclic graphs by giving the max–min characterization results on $\Delta M(G)$ with respect to the number of order and cut edges. The extremal cacti with given parameters concerning $\Delta M(G)$ were obtained in [16]. Much more recently, Wang and Zheng [17] established both sharp lower and upper bounds on $\Delta M(G)$ for maximal plane graphs with minimum degree four and diameter two, and found extremal graphs satisfying these prescribed bounds.

Let $T$ be a tree, in which it no longer has a vertex of degree two, and there exists at least one vertex of degree three or more. A vertex of degree exactly one is a *leaf* of the tree $T$. A tree $T$ is called a *plane tree* if $T$ is a tree that is embedded in the plane. A *Halin graph* is a graph $G = T \cup C$, where $T$ is a plane tree and $C$ is a cycle obtained by connecting all consecutive leaves of $T$ in the cyclic order determined by the embedding of $T$. Sometimes, $T$ is said to be *characteristic tree* of $G$, and $C$ is the *outer cycle* of $G$. The vertices in $V(C)$ and $V(G) \setminus V(C)$ are called *outer vertices* and *inner vertices* of $G$, respectively.

Halin graphs are minimally three-connected plane graphs, that is, they are themselves three-connected, but any of their proper subgraph is not. Bondy and Lovász [18] demonstrated that a Halin graph is almost pancyclic, that is, it has at least one cycle of each length $p$, $3 \le p \le n$, except possibly for one even value of $p$. Particularly, Halin graphs are Hamiltonian. In 2003, Stadler [19] investigated the minimal cycle bases for Halin graphs. Lai et al. [20] established a close relation between the strong chromatic index for a Halin graph and its characteristic tree. Chan et al. [21] showed the edge-face chromatic number of a Halin graph $G$ with $\Delta(G) \ge 5$ which is equal to $\Delta(G)$. Other related property-preserving results on Halin graphs have emerged in [22–24].

The main purpose of this paper is to obtain the sharpness of lower and upper bounds on difference of Zagreb indices concerning Halin graphs by considering two situations: (i) general case; (ii) special case with fewer inner vertices.

## 2. Preliminaries

Assume that $G$ is a Halin graph. Let $V_{\text{inn}}(G)$ and $V_{\text{out}}(G)$ denote the set of inner vertices and outer vertices of $G$, respectively. When $|V_{\text{inn}}(G)| = 1$, $G$ is referred to as a *wheel* with $n$ vertices, denoted $W_n$. Let $v \in V(G)$. We say that $v$ is a *k-vertex* if $d(v) = k$. An inner vertex is called a *handle* if exactly one of its neighbors is an inner vertex. In particular, if $v$ is a handle, and $d(v) = k$, then $v$ is said to be *k-handle*. We define an edge $uv$ of $E(G)$ to be an $(i, j)$-*edge*, where $d(u) = i$ and $d(v) = j$. For simplicity, the set of $(i, j)$-edge, and the number of $(i, j)$-edges of $G$ are denoted by $E_{i,j}$ and $m_{i,j}$, respectively, where $\delta(G) \le i, j \le \Delta(G)$.

**Lemma 1.** *Let $G = T \cup C$ be a Halin graph which is not a wheel. Then $G$ includes at least two handles.*

**Proof.** Note that, because $G$ is not a wheel, $G$ includes at least two inner vertices. Let $F = G - V(C)$. Then, $F$ is a tree with $|V(F)| \ge 2$. Let $P = z_1 z_2 \cdots z_k$ be the longest path in

$F$. According to the longest property of $P$, the assertion that $k \geq 2$ and both $z_1$ and $z_k$ are leaves of $F$ holds. Thus, $z_1$ and $z_k$ are handles of $G$. □

**Lemma 2.** *Let $G = T \cup C$ be a Halin graph on n vertices. Then $|V_{\text{out}}(G)| \geq \lceil \frac{n}{2} \rceil + 1$.*

**Proof.** For $i \geq 1$, let $n_i$ denote the number of $i$-vertices in $T$. Then, $n = n_1 + n_2 + \cdots + n_\Delta$, $n_1 = |V_{\text{out}}(G)|$, where $\Delta = \Delta(T) = \Delta(G)$. Since $T$ has no 2-vertex and $|E(T)| = n - 1$, we obtain:

$$2(n-1) = \sum_{v \in V(T)} d_T(v) = n_1 + \sum_{i=3}^{\Delta} i n_i.$$

If $\Delta \geq 4$, then $2(n-1) \geq n_1 + 4 + 3(n - n_1 - 1)$, so it follows that $n_1 \geq \frac{1}{2}(n+3) \geq \lceil \frac{n}{2} \rceil + 1$. If $\Delta = 3$, then $2(n-1) = n_1 + 3(n - n_1)$, that is, $2n_1 = n + 2$, which implies that $n$ is even, and so $n_1 = \frac{n}{2} + 1 = \lceil \frac{n}{2} \rceil + 1$. □

Lemma 2 asserts that each Halin graph $G$ contains at least three outer vertices. Specifically, if $|V_{\text{out}}(G)| = 3$, then $G \cong K_4$.

**Lemma 3.** *Let $G = T \cup C$ be a Halin graph on $n$ vertices which is not a wheel. Assume that $v_0$ is a $k$-handle with neighbors $u_0, v_1, \ldots, v_{k-1}$ in cyclic order, where $u_0$ is an inner vertex and $v_1, v_2, \ldots, v_{k-1}$ are outer vertices. Assume that $C = v_1 v_2 \cdots v_{k-1} v_k \cdots v_t v_1$, where $t = |V(C)| \geq 4$. Suppose $G' = T' \cup C'$ is a Halin graph obtained from $G$ by carrying out the following operations **(OP1)** and **(OP2)**, as shown in Figure 1:*

**(OP1)** *Contracting the edge $v_0 u_0$ into a vertex $v_0'$, and adding a leaf $u_0'$ at $v_0'$ to form $T'$;*
**(OP2)** *Set $C' = v_1 v_2 \cdots v_{k-1} u_0' v_k \cdots v_t v_1$.*

*Then, $\Delta M(G') > \Delta M(G)$.*

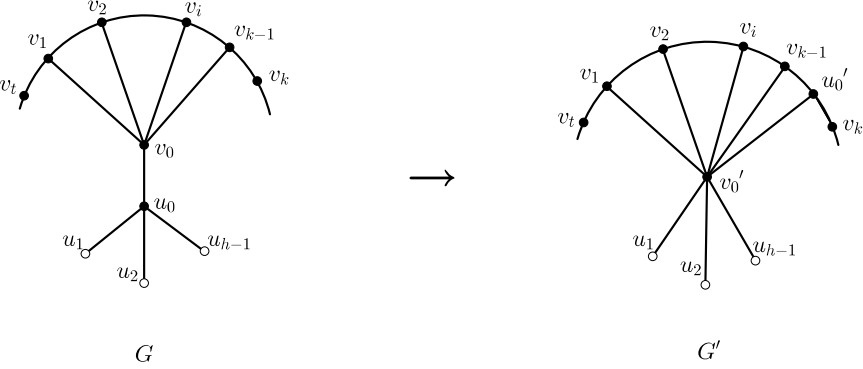

**Figure 1.** Operations of $G$ to $G'$.

**Proof.** It is now easily checked that $G'$ is a Halin graph with $|V(G')| = |V(G)| = n$ and $|E(G')| = |E(G)| + 1$. Suppose that $d_G(u_0) = h$ and $v_0, u_1, u_2, \ldots, u_{h-1}$ are the neighbors of $u_0$ in cyclic order. It follows that $d_{G'}(v_0') = k + h - 1 \geq 3 + 3 - 1 = 5$, $d_{G'}(u_0') = 3$, and for any $x \in V(G') \setminus \{v_0', u_0'\}$, $d_{G'}(x) = d_G(x)$. In particular, for $j = 1, 2, \ldots, h-1$, $d_{G'}(u_j) = d_G(u_j)$. Let

$S = \{e \in E(G) \mid e \text{ is incident with } v_0 \text{ or } u_0\} \cup \{v_{k-1} v_k\}$;
$S' = \{e \in E(G') \mid e \text{ is incident with } v_0'\} \cup \{v_{k-1} u_0', u_0' v_k\}$.

Note that $E(G) - S = E(G') - S'$ and every edge $e \in (E(G) - S) \cap (E(G') - S')$ has the same contribution in both $\Delta M(G)$ and $\Delta M(G')$. For an edge $e = xy \in S$, let

$$\phi(e) = (d_G(x) - 1)(d_G(y) - 1) - 1,$$

and for an edge $e' = x'y' \in S'$, let

$$\phi'(e') = (d_{G'}(x') - 1)(d_{G'}(y') - 1) - 1.$$

To obtain the conclusion, we need to compute the following:

$$
\begin{aligned}
\sigma : \quad &= \sum_{uv \in S} \phi(uv) \\
&= \sum_{uv \in S} [(d_G(u) - 1)(d_G(v) - 1) - 1] \\
&= (k-1)[(3-1)(k-1) - 1] + [(k-1)(h-1) - 1] \\
&+ [(3-1)(3-1) - 1] + [(h-1)(d_G(u_1) - 1) - 1] \\
&+ [(h-1)(d_G(u_2) - 1) - 1] + \cdots + [(h-1)(d_G(u_{h-1}) - 1) - 1],
\end{aligned}
$$

and

$$
\begin{aligned}
\sigma' : \quad &= \sum_{uv \in S'} \phi'(uv) \\
&= \sum_{uv \in S'} [(d_{G'}(u) - 1)(d_{G'}(v) - 1) - 1] \\
&= k[(3-1)(k + h - 1 - 1) - 1] + 2[(3-1)(3-1) - 1] \\
&+ [(k + h - 2)(d_G(u_1) - 1) - 1] + [(k + h - 2)(d_G(u_2) - 1) - 1] \\
&+ \cdots + [(k + h - 2)(d_G(u_{h-1}) - 1) - 1].
\end{aligned}
$$

Since

$$
\begin{aligned}
\sigma' - \sigma &= kh + k + h + (k-1)[d_G(u_1) + d_G(u_2) + \cdots + d_G(u_{h-1}) - (h-1)] \\
&\geq kh + k + h + 2(k-1)(h-1) \\
&= 3kh - k - h + 2 \\
&= (k-1)(h-1) + 2kh + 1 \\
&\geq (3-1)(3-1) + 2 \times 3 \times 3 + 1 = 23,
\end{aligned}
$$

we derive

$$
\Delta M(G') - \Delta M(G) = \sigma' - \sigma \geq 23 > 0.
$$

$\square$

A direct consequence of Lemma 3 is the next simple lemma.

**Lemma 4.** *Let G be a Halin graph on n vertices and $|V_{\mathrm{inn}}(G)| \geq 2$. Then, repeating the above operations $|V_{\mathrm{inn}}(G)| - 1$ times, we obtain finally a wheel $W_n$.*

With an easy computation, we can obtain the next lemma:

**Lemma 5.** *For $n \geq 3$, $\Delta M(W_n) = 2(n-1)^2$.*

### 3. General Halin Graphs

In this section, we present tight lower and upper bounds on difference of Zagreb indices for any Halin graphs, and characterize corresponding extremal graphs.

**Theorem 1.** *Let G be a Halin graph on n vertices. Then $\Delta M(G) \leq 2(n-1)^2$, where the equality holds if and only if $G \cong W_n$.*

**Proof.** If $G$ is a wheel, the result deduces immediately from Lemma 5. Thus, below, we assume that $G$ is a nonwheel. Let $k = |V_{\mathrm{inn}}(G)| \geq 2$. By Lemma 4, by repeating the above operations $k - 1$ times, we obtain a sequence of graphs $G_0, G_1, \ldots, G_{k-1}$, where $G = G_0$, and $G_{k-1} \cong W_n$. By Lemma 3, for any $0 \leq i \leq k - 1$, we know that $\Delta M(G_{i+1}) - \Delta M(G_i) \geq 23$. Hence,

$$
\Delta M(G_0) < \Delta M(G_1) < \cdots < \Delta M(G_{k-1}) = 2(n-1)^2.
$$

This shows that $\Delta M(G) < 2(n-1)^2$ if $G$ is a nonwheel. Hence, the result establishes the above. $\square$

A Halin graph with $n$ vertices is *special* if it contains one 4-vertex and $(n-1)$ 3-vertices.

**Lemma 6.** *Suppose $G$ is a Halin graph with $n$ vertices and $m$ edges.*

(1) *If $n$ is even, then $m \geq \frac{3}{2}n$, where the equality attains if and only if $G$ is 3-regular.*
(2) *If $n$ is odd, then $m \geq \frac{1}{2}(3n+1)$, where the equality attains if and only if $G$ is special.*

**Proof.** Note that $\delta(G) = 3$ and $\Delta(G) \geq 3$. If $n$ is even, then since

$$2m = \sum_{v \in V(G)} d(v) \geq 3n,$$

it follows that $m \geq \frac{3}{2}n$. Obviously, equality of the lower bound is attained if and only if $G$ is a 3-regular Halin graph. Assume that $n$ is odd. Since there is no 3-regular graph of odd order, it yields that $\Delta(G) \geq 4$ and the following expressions hold:

$$2m = \sum_{v \in V(G)} d(v) \geq 4 + 3(n-1) = 3n+1.$$

Therefore, $m \geq \frac{1}{2}(3n+1)$. Similarly, equality of the lower bound is attained if and only if $G$ is a special Halin graph. $\square$

It is not hard to check that there exist only one 3-regular Halin graph on four vertices (i.e., $K_4$), one 3-regular Halin graph on six vertices (i.e., the triangular prism), one 3-regular Halin graph on eight vertices, and three 3-regular Halin graphs on ten vertices. These graphs are depicted in Figure 2.

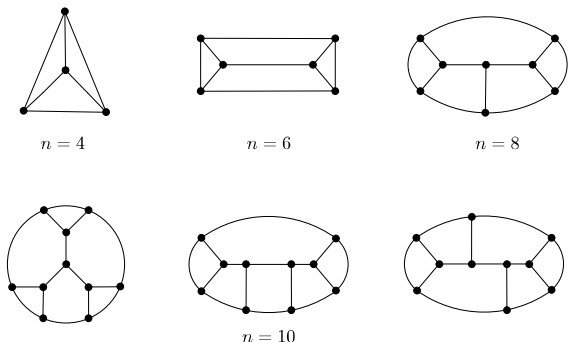

**Figure 2.** 3-regular Halin graphs with $n = 4, 6, 8, 10$.

Analogously, there exist only one special Halin graph on five vertices (i.e., $W_5$), one special Halin graph on seven vertices, three special Halin graphs on nine vertices, and eight special Halin graphs on eleven vertices. These graphs are depicted in Figure 3.

**Lemma 7.** *Let $n \geq 4$ be an integer.*

(1) *If $n$ is even, then there is a 3-regular Halin graph on $n$ vertices.*
(2) *If $n$ is odd, then there is a special Halin graph on $n$ vertices.*

**Proof.** (1) If $n = 4, 6, 8, 10$, then the result holds automatically by the foregoing analysis and Figure 2. Assume that $n \geq 12$ is even. Let $P = x_1 x_2 \cdots x_k$ be a path on $k \geq 5$ vertices. Let $T$ be a plane tree constructed from $P$ by adding two leaves at each of $x_1$ and $x_k$ and one leaf at each of $x_2, x_3, \ldots, x_{k-1}$. Let $G$ denote the Halin graph with $T$ as its characteristic tree. Then $n = 2k + 2 \geq 12$ is even and $G$ is our required graph.

(2) If $n = 5, 7, 9, 11$, then the result is true by Figure 3. Assume that $n \geq 13$ is odd. Let $P = x_1 x_2 \cdots x_k$ be a path on $k \geq 5$ vertices. Let $T$ be a plane tree constructed from $P$ by adding three leaves at $x_1$, two leaves in $x_k$, and one leaf at each of $x_2, x_3, \ldots, x_{k-1}$. Let $G$ denote the Halin graph with $T$ as its characteristic tree. Then $n = 2k + 3 \geq 13$ is odd and $G$ is our required graph. $\square$

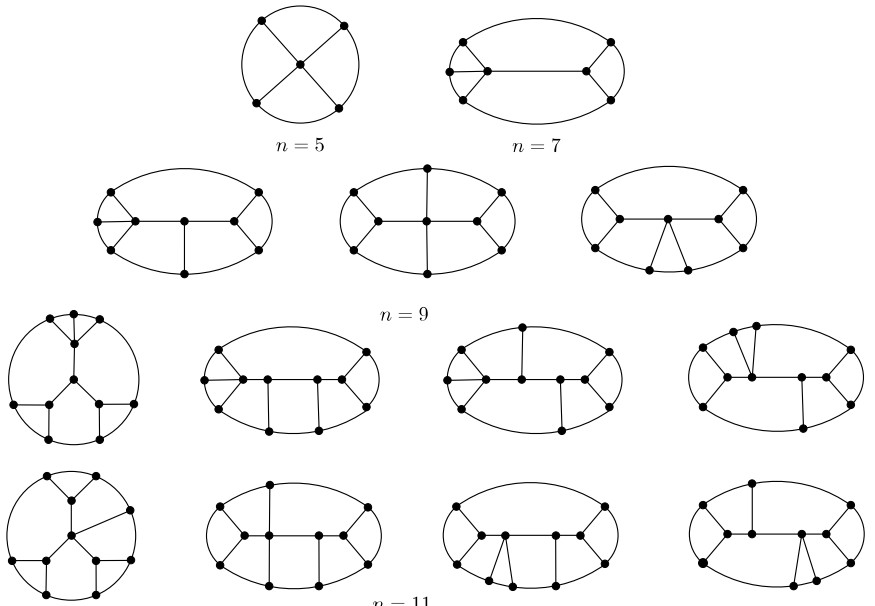

**Figure 3.** Special Halin graphs with $n = 5, 7, 9, 11$.

By a simple computation, we immediately derive the next lemma:

**Lemma 8.** *Let $G$ be a 3-regular Halin graph with $n \geq 4$ being even. Then $\Delta M(G) = \frac{9}{2}n$.*

**Lemma 9.** *Let $G$ be a special Halin graph with $n \geq 5$ being odd. Then $\Delta M(G) = \frac{9}{2}n + \frac{19}{2}$.*

**Proof.** By Lemma 6(2), $m = \frac{1}{2}(3n + 1)$. Since $V(G)$ consists of one 4-vertex and $(n-1)$ 3-vertices, we have that $E(G) = E_{3,3} \cup E_{3,4}$, $m_{3,3} = m - 4$, and $m_{3,4} = 4$. Thus,

$$
\begin{aligned}
\Delta M(G) &= \sum_{uv \in E(G)} (d(u) - 1)(d(v) - 1) - m \\
&= 4(4 - 1)(3 - 1) + (m - 4)(3 - 1)(3 - 1) - m \\
&= 3m + 8 \\
&= 3\left[\frac{1}{2}(3n + 1)\right] + 8 \\
&= \frac{9}{2}n + \frac{19}{2}.
\end{aligned}
$$

$\square$

Now we determine a tight lower bound on $\Delta M(G)$ for a Halin graph $G$ with $n$ vertices. If $n = 4$, then $G \cong W_4$, and $\Delta M(W_4) = 18$ by Lemma 8. If $n = 5$, then $G \cong W_5$, and $\Delta M(W_5) = 32$ by Lemma 9. In general, for $n \geq 6$, we have the following:

**Theorem 2.** *Let $G = T \cup C$ be a Halin graph on $n \geq 6$ vertices. Then the next results are presented.*

*(1)  If $n$ is even, then $\Delta M(G) \geq \frac{9n}{2}$, where the equality holds if and only if $G$ is 3-regular.*
*(2)  If $n$ is odd, then $\Delta M(G) \geq \frac{9n}{2} + \frac{19}{2}$, where the equality holds if and only if $G$ is special.*

**Proof.** Using Lemmas 6(1), 7(1), and 8, we can infer the result (1). Similarly, by Lemmas 6(2), 7(2), and 9, the result (2) holds automatically. □

### 4. Halin Graphs with Fewer Inner Vertices

Given an integer $k \geq 1$, let $\mathcal{H}_n^k$ denote the class of all Halin graphs with $n$ vertices and $k$ inner vertices. In particular, if $k = 1$, then $\mathcal{H}_n^1 = \{W_n\}$. This section is dedicated to investigating the difference of Zagreb indices regarding Halin graphs having fewer inner vertices.

#### 4.1. Halin Graphs with Two Inner Vertices

Suppose $G = T \cup C \in \mathcal{H}_n^2$ is a Halin graph. Then $|V_{\text{inn}}(G)| = 2$. Assume that $V(G) = \{u, v; u_1, \ldots, u_{p-1}; v_1, \ldots, v_{q-1}\}$, $C = u_1 u_2 \cdots u_{p-1} v_1 v_2 \cdots v_{q-1} u_1$, $E(G) = E(C) \cup \{uv; uu_1, \ldots, uu_{p-1}; vv_1, \ldots, vv_{q-1}\}$, where $d(u) = p \geq 3$, $d(v) = q \geq 3$, and $n = p + q$, as illustrated in Figure 4.

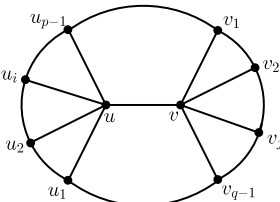

**Figure 4.** $G \in \mathcal{H}_n^2$ with $n = p + q$.

**Theorem 3.** *Let $G = T \cup C \in \mathcal{H}_n^2$ be a Halin graph with $n = p + q$ and $q \geq p$, which is shown in Figure 4. Then the next results can be established:*

*(1)　$\Delta M(G) \leq 2n^2 - 12n + 27$, where the equality holds for $p = 3$ and $q = n - 3$.*

*(2)　If $n$ is even, then $\Delta M(G) \geq \frac{5}{4}n^2 - 3n$, where the equality holds for $p = q = \frac{n}{2}$; If $n$ is odd, then $\Delta M(G) \geq \frac{5}{4}n^2 - 3n + \frac{3}{4}$, where the equality holds for $p = \frac{n-1}{2}$ and $q = \frac{n+1}{2}$.*

**Proof.** The definition implies that $|V_{\text{inn}}(G)| = 2$, $|V(C)| = n - 2$, and $m = (n - 1) + (n - 2) = 2n - 3$. Note that $G$ has only $(3,3)$-edges, $(3,p)$-edges, $(3,q)$-edges, and $(p,q)$-edge. Furthermore, observe that $m_{3,3} = n - 2$, $m_{3,p} = p - 1$, $m_{3,q} = q - 1$, $m_{p,q} = 1$. Since $q = n - p$, we have the following:

$$
\begin{aligned}
\Delta M(G) &= \sum_{uv \in E(G)} (d(u) - 1)(d(v) - 1) - m \\
&= (3 - 1)(3 - 1)(n - 2) + (3 - 1)(p - 1)(p - 1) \\
&+ (3 - 1)(q - 1)(q - 1) + (p - 1)(q - 1) - (2n - 3) \\
&= 2n^2 - 3n + 3p^2 - 3pn.
\end{aligned}
$$

Let

$$
f(p) = 3p^2 - 3pn.
$$

It is actually necessary to view $f(p)$ as a continuous function of $p$ to search for the minimum and maximum values of $f(p)$. Evidently, the function $f(p)$ decreases strictly monotonically for $p \in [3, \lfloor \frac{n}{2} \rfloor]$. Consequently, its maximum value achieves at $p = 3$:

$$
f(3) = 3 \times 3^2 - 3 \times 3n = 27 - 9n.
$$

This implies that $\Delta M(G)$ achieves its maximum value:

$$
\Delta M(G) = 2n^2 - 3n + 27 - 9n = 2n^2 - 12n + 27.
$$

On the other hand, it is easy to show that $f(p)$ obtains its minimum value if $p = \lfloor \frac{n}{2} \rfloor$. Precisely, if $n$ is even, then $p = \frac{n}{2}$, and hence $f(\frac{n}{2}) = 3(\frac{n}{2})^2 - 3n \times \frac{n}{2} = -\frac{3}{4}n^2$. Consequently, $\Delta M(G)$ obtains its minimum value:

$$\Delta M(G) = 2n^2 - 3n - \frac{3}{4}n^2 = \frac{5}{4}n^2 - 3n.$$

If $n$ is odd, then $p = \frac{n-1}{2}$, and hence $f(\frac{n-1}{2}) = 3(\frac{n-1}{2})^2 - 3n \times \frac{n-1}{2} = -\frac{3}{4}n^2 + \frac{3}{4}$. It turns out that $\Delta M(G)$ attains its minimum value:

$$\Delta M(G) = 2n^2 - 3n - \frac{3}{4}n^2 + \frac{3}{4} = \frac{5}{4}n^2 - 3n + \frac{3}{4}.$$

□

### 4.2. Halin Graphs with Three Inner Vertices

Let $G = T \cup C \in \mathcal{H}_n^3$ be a Halin graph with $V_{\text{inn}}(G) = \{u, v, w\}$. Then $\{u, v, w\}$ forms a path in $T$, say $uvw$. Assume that $d(u) = p$, $d(v) = q$, and $d(w) = r$. Then $p + q + r = n + 1$, $|V(C)| = n - 3$, and $m = (n-1) + (n-3) = 2n - 4$, where $p, q, r \geq 3$.

**Lemma 10.** *Let $G \in \mathcal{H}_n^3$ be a Halin graph defined above. Then*

$$\Delta M(G) = 2p^2 + q^2 + 2r^2 + (n-2)q - 3(n+1).$$

**Proof.** Since $n = p + q + r - 1$, $m = 2n - 4$, $E(G) = E(C) \cup E_{3,p} \cup E_{3,q} \cup E_{3,r} \cup E_{p,q} \cup E_{q,r}$, $|E(C)| = n - 3$, $m_{3,p} = p - 1$, $m_{3,q} = q - 2$, $m_{3,r} = r - 1$, $m_{p,q} = 1$, and $m_{q,r} = 1$, we have the following:

$$\begin{aligned}
\Delta M(G) &= \sum_{uv \in E(G)} (d(u) - 1)(d(v) - 1) - m \\
&= (3-1)(3-1)(n-3) + (3-1)(p-1)(p-1) + (3-1)(q-1)(q-2) \\
&\quad + (3-1)(r-1)(r-1) + (p-1)(q-1) + (q-1)(r-1) - (2n-4) \\
&= 2p^2 + 2q^2 + 2r^2 - 3q + pq + qr - 3(n+1) \\
&= 2p^2 + q^2 + 2r^2 + (n-2)q - 3(n+1).
\end{aligned}$$

□

**Lemma 11.** *Let $G = T \cup C \in \mathcal{H}_n^3$ be a Halin graph with $p \geq r \geq 4$ defined above. Let $T'$ denote a plane tree obtained from $T$ by removing a leaf at $w$ and then adding a leaf at $u$. Let $G' = T' \cup C'$ be a Halin graph with $T'$ as characteristic tree. Then $\Delta M(G') > \Delta M(G)$.*

**Proof.** Note that $G' \in \mathcal{H}_n^3$ satisfies that $|V(G')| = |V(G)| = n$, $|E(G')| = |E(G)| = m$, $|V(C')| = |V(C)|$, $V_{\text{inn}}(G') = V_{\text{inn}}(G) = \{u, v, w\}$, $d_{G'}(u) = d_G(u) + 1 = p + 1$, $d_{G'}(v) = d_G(v) = q$, $d_{G'}(w) = d_G(w) - 1 = r - 1 \geq 3$. By Lemma 10,

$$\begin{aligned}
\Delta M(G') - \Delta M(G) &= 2(p+1)^2 + q^2 + 2(r-1)^2 + (n-2)q - 3(n+1) \\
&\quad - [2p^2 + q^2 + 2r^2 + (n-2)q - 3(n+1)] \\
&= 4(p-r) + 4 \geq 4.
\end{aligned}$$

□

Lemma 11 tells us that it is sufficient to consider the case that $p \geq r = 3$ to maximize the difference of Zagreb indices for the graphs in $\mathcal{H}_n^3$.

**Theorem 4.** *Let $G \in \mathcal{H}_n^3$ be a Halin graph with $r = 3$. Then $\Delta M(G) \leq 2n^2 - 20n + 68$, with equality if and only if the following statements* (1) *and* (2) *hold:*

(1)    If $q \geq p$, then $p = 3$ and $q = n - 5$;
(2)    If $p > q$, then $q = 3$ and $p = n - 5$.

**Proof.** Since $r = 3$, we have $p + q = n + 1 - 3 = n - 2$. By Lemma 10, $\Delta M(G) = 2p^2 + 2q^2 + pq - 3(n - 5)$. Let

$$f(p, q) = 2p^2 + 2q^2 + pq - 3(n - 5).$$

(1) Suppose that $q \geq p$. Our aim is to maximize $f(p, q)$, based on the conditions that $q \geq p \geq 3$ and $p + q = n - 2$. In fact, when $p \geq 4$, we obtain:

$$\begin{aligned}
f(p - 1, q + 1) - f(p, q) &= 2(p-1)^2 + 2(q+1)^2 + (p-1)(q+1) - 3(n-5) \\
&\quad - [2p^2 + 2q^2 + pq - 3(n-5)] \\
&= 3(q - p) + 3 \geq 3.
\end{aligned}$$

The above inequality indicates that the function $f(p, q)$ increases strictly monotonically while increasing $q$ and decreasing $p$ simultaneously. Hence, $f(p, q)$ achieves its maximum value at $(3, n - 5)$:

$$f(3, n - 5) = 2 \times 3^2 + 2(n - 5)^2 + 3(n - 5) - 3(n - 5) = 2n^2 - 20n + 68.$$

This implies that $\Delta M(G)$ attains its maximum value $2n^2 - 20n + 68$ at $(3, n - 5, 3)$.

(2) Suppose that $p > q$. Similarly, our aim is also to maximize $f(p, q)$ that satisfies $p > q \geq 3$ and $p + q = n - 2$. If $q \geq 4$, then

$$\begin{aligned}
f(p + 1, q - 1) - f(p, q) &= 2(p+1)^2 + 2(q-1)^2 + (p+1)(q-1) - 3(n-5) \\
&\quad - [2p^2 + 2q^2 + pq - 3(n-5)] \\
&= 3(p - q) + 3 \geq 3.
\end{aligned}$$

This confirms that the function $f(p, q)$ increases strictly monotonically as $p$ increases and $q$ decreases simultaneously, therefore it achieves the maximum value at $(n - 5, 3)$:

$$f(n - 5, 3) = 2(n - 5)^2 + 2 \times 3^2 + 3(n - 5) - 3(n - 5) = 2n^2 - 20n + 68.$$

It follows that $\Delta M(G)$ attains its maximum value $2n^2 - 20n + 68$ at $(n - 5, 3, 3)$.    □

When $n = 12$, an easy calculation can be used to find exactly four Halin graphs, depicted in Figure 5, which attain the maximum value of the difference of Zagreb indices in $\mathcal{H}_{12}^3$.

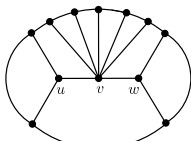 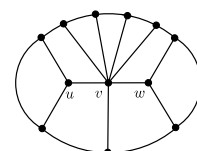 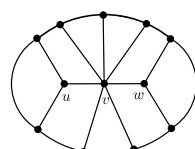 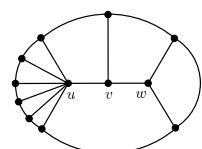

**Figure 5.** Graphs attaining the maximum value of the difference of Zagreb indices in $\mathcal{H}_{12}^3$.

**Lemma 12.** *Let $G = T \cup C \in \mathcal{H}_n^3$ be a Halin graph defined above such that $p \geq r + 2 \geq 5$. Let $T'$ denote a plane tree obtained from $T$ by removing a leaf at $u$ and then adding a leaf at $w$. Let $G' = T' \cup C'$ be a Halin graph with $T'$ as characteristic tree. Then $\Delta M(G') < \Delta M(G)$.*

**Proof.** Note that $G' \in \mathcal{H}_n^3$ satisfies that $|V(G')| = |V(G)| = n$, $|E(G')| = |E(G)| = m$, $|V(C')| = |V(C)|$, $V_{\text{inn}}(G') = V_{\text{inn}}(G) = \{u, v, w\}$, $d_{G'}(u) = d_G(u) - 1 = p - 1 \geq 4$, $d_{G'}(v) = d_G(v) = q$, and $d_{G'}(w) = d_G(w) + 1 = r + 1$. By Lemma 10,

$$\begin{aligned}
\Delta M(G') - \Delta M(G) &= 2(p-1)^2 + q^2 + 2(r+1)^2 + (n-2)q - 3(n+1) \\
&\quad - [2p^2 + q^2 + 2r^2 + (n-2)q - 3(n+1)] \\
&= -4(p-r) + 4 \le -4.
\end{aligned}$$

□

Lemma 12 is used to provide the condition that suffices for handling the cases $p = r$ and $p = r + 1$ to minimize the difference of Zagreb indices for graphs in $\mathcal{H}_n^3$.

**Theorem 5.** *Let $G \in \mathcal{H}_n^3$ be a Halin graph with $p = r$ or $p = r + 1$, and $\eta(n) = \frac{7n^2}{8} - 2n - 3$. Then $\Delta M(G) \ge \eta(n) - \epsilon$, where*

*(1)　If $n \equiv 0 \pmod 8$, then $\epsilon = 1$;*
*(2)　If $n \equiv 1, 7 \pmod 8$, then $\epsilon = -\frac{1}{8}$;*
*(3)　If $n \equiv 2, 6 \pmod 8$, then $\epsilon = \frac{1}{2}$;*
*(4)　If $n \equiv 3, 5 \pmod 8$, then $\epsilon = \frac{7}{8}$;*
*(5)　If $n \equiv 4 \pmod 8$, then $\epsilon = 0$.*

*Furthermore, we determine extremal graphs with these lower bounds.*

**Proof.** Note that $p + r = n + 1 - q$, and $p, q, r \ge 3$. If $n, q$ are both odd or even, then $p = r + 1$; Otherwise, $p = r$.

● Assume that $p = r$. Then $2r + q = n + 1$ and $3 \le q = n + 1 - 2r \le n + 1 - 6 = n - 5$. By Lemma 10, $\Delta M(G) = 4r^2 + q^2 + (n-2)q - 3(n+1) = 2q^2 - (n+4)q + n^2 - n - 2$. Let

$$f(q) = 2q^2 - (n+4)q = 2(q - \frac{n+4}{4})^2 - \frac{(n+4)^2}{8}.$$

The objective is to minimize the continuous function $f(q)$, based on the condition that $q$ is a variable. Thanks to $3 \le q \le n - 5$, $f(q)$ achieves the minimum value if $q = \lfloor \frac{n+4}{4} \rfloor$.

● Assume that $p = r + 1$. Then $2r + q + 1 = n + 1$ and $3 \le q = n - 2r \le n - 6$. By Lemma 10, $\Delta M(G) = (2r+1)^2 + q^2 + (n-2)q - 3n - 2 = 2q^2 - (n+4)q + n^2 - n - 1$. Let

$$f(q) = 2q^2 - (n+4)q = 2(q - \frac{n+4}{4})^2 - \frac{(n+4)^2}{8}.$$

Since $3 \le q \le n - 6$, $f(q)$ attains its minimum value at $q = \lfloor \frac{n+4}{4} \rfloor$.

Now, according to the size of $n$, we split the proof into eight subcases as follows. Let $\Delta^*$ denote the minimum value of $\Delta M(G)$ in every possible case.

(1) $n \equiv 0 \pmod 8$. Then $n$ is even, $q = \frac{n+4}{4}$ is odd, $p = r = \frac{3n}{8}$, and

$$f(\frac{n+4}{4}) = -\frac{(n+4)^2}{8} = -\frac{n^2}{8} - n - 2.$$

$\Delta M(G)$ attains its minimum value:

$$\Delta^* = -\frac{n^2}{8} - n - 2 + n^2 - n - 2 = \frac{7n^2}{8} - 2n - 4.$$

Consequently, we have $\Delta M(G) \ge \Delta^* = \frac{7n^2}{8} - 2n - 4$.

(2) $n \equiv 1 \pmod 8$. Then $n$ is odd, $q = \frac{n+3}{4}$ or $q = \frac{n+7}{4}$. If $q = \frac{n+3}{4}$, then $q$ is odd, which implies that $p = r + 1$, $p = \frac{3n+5}{8}$, $r = \frac{3n-3}{8}$, and $\Delta M(G)$ obtaining the minimum value:

$$\Delta^* = 2(\frac{n+3}{4})^2 - (n+4)(\frac{n+3}{4}) + n^2 - n - 1 = \frac{7n^2}{8} - 2n - \frac{23}{8}.$$

If $q = \frac{n+7}{4}$, then $q$ is even, which implies that $p = r = \frac{3n-3}{8}$, and the minimum value of $\Delta M(G)$ is as follows:

$$\Delta^* = 2(\frac{n+7}{4})^2 - (n+4)(\frac{n+7}{4}) + n^2 - n - 2 = \frac{7n^2}{8} - 2n - \frac{23}{8}.$$

Consequently, we always have $\Delta M(G) \geq \Delta^* = \frac{7n^2}{8} - 2n - \frac{23}{8}$.

(3) $n \equiv 2 \pmod 8$. Then $n$ is even, $q = \frac{n+2}{4}$ or $q = \frac{n+6}{4}$. If $q = \frac{n+2}{4}$, then $q$ is odd, which implies that $p = r = \frac{3n+2}{8}$, and $\Delta M(G)$ attains its minimum value:

$$\Delta^* = 2(\frac{n+2}{4})^2 - (n+4)(\frac{n+2}{4}) + n^2 - n - 2 = \frac{7n^2}{8} - 2n - \frac{7}{2}.$$

If $q = \frac{n+6}{4}$, then $q$ is even, which implies that $p = r + 1$, $p = \frac{3n+2}{8}$, $r = \frac{3n-6}{8}$, and $\Delta M(G)$ attains its minimum value:

$$\Delta^* = 2(\frac{n+6}{4})^2 - (n+4)(\frac{n+6}{4}) + n^2 - n - 1 = \frac{7n^2}{8} - 2n - \frac{5}{2}.$$

Consequently, we always have $\Delta M(G) \geq \frac{7n^2}{8} - 2n - \frac{7}{2}$.

(4) $n \equiv 3 \pmod 8$. Then $n$ is odd, $q = \frac{n+1}{4}$ or $q = \frac{n+5}{4}$. If $q = \frac{n+1}{4}$, then $q$ is odd, which implies that $p = r + 1$, $p = \frac{3n+7}{8}$, $r = \frac{3n-1}{8}$, and $\Delta M(G)$ attains its minimum value:

$$\Delta^* = 2(\frac{n+1}{4})^2 - (n+4)(\frac{n+1}{4}) + n^2 - n - 1 = \frac{7n^2}{8} - 2n - \frac{15}{8}.$$

If $q = \frac{n+5}{4}$, then $q$ is even, $p = r = \frac{3n-1}{8}$, and $\Delta M(G)$ attains its minimum value:

$$\Delta^* = 2(\frac{n+5}{4})^2 - (n+4)(\frac{n+5}{4}) + n^2 - n - 2 = \frac{7n^2}{8} - 2n - \frac{31}{8}.$$

Thus, it always holds that $\Delta M(G) \geq \frac{7n^2}{8} - 2n - \frac{31}{8}$.

(5) $n \equiv 4 \pmod 8$. Then $n$ is even, $q = \frac{n+4}{4}$, $r = \frac{3n-4}{8}$, $p = \frac{3n+4}{8}$, and $\Delta M(G)$ attains its minimum value:

$$\Delta^* = 2(\frac{n+4}{4})^2 - (n+4)(\frac{n+4}{4}) + n^2 - n - 1 = \frac{7n^2}{8} - 2n - 3.$$

Thus, it always holds that $\Delta M(G) \geq \frac{7n^2}{8} - 2n - 3$.

(6) $n \equiv 5 \pmod 8$. Then $n$ is odd, $q = \frac{n+3}{4}$ or $q = \frac{n+7}{4}$. If $q = \frac{n+3}{4}$, then $q$ is even, which implies that $p = r = \frac{3n+1}{8}$, and $\Delta M(G)$ attains its minimum value:

$$\Delta^* = 2(\frac{n+3}{4})^2 - (n+4)(\frac{n+3}{4}) + n^2 - n - 2 = \frac{7n^2}{8} - 2n - \frac{31}{8}.$$

If $q = \frac{n+7}{4}$, then $q$ is odd, which implies that $p = r + 1$, $p = \frac{3n+1}{8}$, $r = \frac{3n-7}{8}$, and $\Delta M(G)$ obtains the minimum value:

$$\Delta^* = 2(\frac{n+7}{4})^2 - (n+4)(\frac{n+7}{4}) + n^2 - n - 1 = \frac{7n^2}{8} - 2n - \frac{15}{8}.$$

Thus, it always holds that $\Delta M(G) \geq \frac{7n^2}{8} - 2n - \frac{31}{8}$.

(7) $n \equiv 6 \pmod 8$. Then $n$ is even, $q = \frac{n+2}{4}$ or $q = \frac{n+6}{4}$. If $q = \frac{n+2}{4}$, then $q$ is even, which implies that $p = r + 1$, $p = \frac{3n+6}{8}$, $r = \frac{3n-2}{8}$, and $\Delta M(G)$ attains its minimum value:

$$\Delta^* = 2(\frac{n+2}{4})^2 - (n+4)(\frac{n+2}{4}) + n^2 - n - 1 = \frac{7n^2}{8} - 2n - \frac{5}{2}.$$

If $q = \frac{n+6}{4}$, then $q$ is odd, which implies that $p = r = \frac{3n-2}{8}$, and $\Delta M(G)$ attains its minimum value:

$$\Delta^* = 2(\frac{n+6}{4})^2 - (n+4)(\frac{n+6}{4}) + n^2 - n - 2 = \frac{7n^2}{8} - 2n - \frac{7}{2}.$$

Consequently, we always have $\Delta M(G) \geq \frac{7n^2}{8} - 2n - \frac{7}{2}$.

(8) $n \equiv 7 \pmod{8}$. Then $n$ is odd, $q = \frac{n+1}{4}$ or $q = \frac{n+5}{4}$. If $q = \frac{n+1}{4}$, then $q$ is even, which implies that $p = r = \frac{3n+3}{8}$, and $\Delta M(G)$ attains its minimum value:

$$\Delta^* = 2(\frac{n+1}{4})^2 - (n+4)(\frac{n+1}{4}) + n^2 - n - 2 = \frac{7n^2}{8} - 2n - \frac{23}{8}.$$

If $q = \frac{n+5}{4}$, then $q$ is odd, which implies that $p = r + 1$, $p = \frac{3n+3}{8}$, $r = \frac{3n-5}{8}$, and $\Delta M(G)$ attains its minimum value:

$$\Delta^* = 2(\frac{n+5}{4})^2 - (n+4)(\frac{n+5}{4}) + n^2 - n - 1 = \frac{7n^2}{8} - 2n - \frac{23}{8}.$$

Thus, it always holds that $\Delta M(G) \geq \frac{7n^2}{8} - 2n - \frac{23}{8}$.

Hence, by the foregoing discussion, we summarize that $\Delta M(G) \geq \frac{7n^2}{8} - 2n - 4$ if $n \equiv 0 \pmod{8}$; $\Delta M(G) \geq \frac{7n^2}{8} - 2n - \frac{23}{8}$ if $n \equiv 1, 7 \pmod{8}$; $\Delta M(G) \geq \frac{7n^2}{8} - 2n - \frac{7}{2}$ if $n \equiv 2, 6 \pmod{8}$; $\Delta M(G) \geq \frac{7n^2}{8} - 2n - \frac{31}{8}$ if $n \equiv 3, 5 \pmod{8}$; $\Delta M(G) \geq \frac{7n^2}{8} - 2n - 3$ if $n \equiv 4 \pmod{8}$. In addition, we also find corresponding extremal graphs with the sharpness of lower bounds. $\square$

## 5. Conclusions

In this paper, we determined sharpnessabove these bounds on the difference of Zagreb indices for general Halin graphs. Furthermore, we obtained extremal values on the difference of Zagreb indices for Halin graphs with the number of inner vertices at most three by Theorems 3–5. All corresponding graphs for extremal values were completely found.

In conclusion, it is natural to raise the following question:

**Problem 1.** *For $k = 4, 5$, establish both lower and upper bounds on the difference of Zagreb indices for Halin graphs in $\mathcal{H}_n^k$, and find corresponding extremal graphs.*

**Author Contributions:** Writing—original draft preparation, L.Z.; formal analysis, Y.W.; writing—review and editing, W.W. All authors have read and agreed to the published version of the manuscript.

**Funding:** This research was funded by the NSFC (No. 12161094), NSFC (No. 12071048, 12161141006), NSFC (No. 12031018, 12226303), Science and Technology Commission of Shanghai Municipality (No. 18dz2271000).

**Institutional Review Board Statement:** Not applicable.

**Informed Consent Statement:** Not applicable.

**Data Availability Statement:** All relevant data are provided within the paper.

**Conflicts of Interest:** The authors declare no conflicts of interest.

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
