# Peer review of "The Difference of Zagreb Indices of Halin Graphs"

_axioms, doi:10.3390/axioms12050450_

Round 1

Author Response

Dear the reviewer,

Sorry, I sent the wrong attachment before.  Please see the latest attachment.

Best regards,

Lina Zheng

Reviewer 3 Report

See the attached file, for the improvement of this research work.

Reviewer 4 Report

My remarks are included in a separated file.

Author Response

Dear the reviewer,

Best regards, 

Lina Zheng

Reviewer 5 Report

The report of the paper:

The difference of Zagreb indices of Halin graphs

by Lina Zheng, Yiqiao Wang and Weifan Wang.

In the paper the authors study the difference of Zagreb indices in some Halin graphs. In particular they obtain the lower and upper bounds on the diference Zagreb indices in Halin graphs and they give a characterization of extremal graphs achieving the extremal values.

The topic considered in that paper is classical and seems to be interesting for readers which interested in topological indices in graphs. Genarally the paper is well written, proofs seems be correct. However the manuscript should be inspect once more and some sentences such for example: page1. Line 12: „Usually we set n=|V(G)| and m=|E(G)|” should be reformulated.

I recommend publishing the paper in Axioms.

Author Response

(The authors gave the same response as above.)

Round 2

Reviewer 1 Report

Dear authors, in my previous report, I brought attention to the high similarity index of 41% in your paper and the potential for plagiarism. Despite my recommendation for a thorough review of the paper to ensure proper citation and indication of any borrowed text, it appears that the revised version still has a similar issue with a similarity index of 40%. I consider this to be a direct plagiarism, even if the text is borrowed from your own previous work. I urge you to take immediate action to correct this major issue and address the plagiarism concerns. Failure to do so may result in the rejection of the paper.

Author Response

 Dear the editor,

According to the referee’s comments, we have made a major revision for our paper (Manuscript Number: Axioms-2169039), entitled with “The difference of Zagreb indices of Halin graphs” . The previous submitted version is somewhat repetitive with our own previous work, so we have made major changes to the content of our paper to reduce the repetition rate.

Best regards,

Lina Zheng

School of Mathematical Sciences

Zhejiang Normal University

Jinhua 321004, China

Round 3

Reviewer 1 Report

The quality of the revised version is way better than the previous version. I can now recommend the publication.